


# The role of citizen science to assess the spatiotemporal pattern of rainfall events in urban areas: a case study in the city of Genoa, Italy

Nicola Loglisci[1], Giorgio Boni[2], Arianna Cauteruccio[2], Francesco Faccini[3,4], Massimo Milelli[1], Guido Paliaga[5], and Antonio Parodi[1]

[1]CIMA Research Foundation, Savona, Italy
[2]DICCA, University of Genova, Italy
[3]DISTAV, University of Genova, Italy
[4]CNR-IRPI, Torino, Italy
[5]GISIG – Geographical Information Systems International Group, Genova, Italy

**Correspondence:** Francesco Faccini (faccini@unige.it)

**Abstract.** Climate change in the Mediterranean region is evidenced by an increase in average air temperature and a variation in rainfall regime: the value of cumulated annual rainfall seems to be basically constant, however, rainfall of maximum intensity and short duration, between 1 and 24 hours, is increasing, especially in the period between late summer and early autumn. The associated ground effects in urban areas consist of flash floods and pluvial floods, often in very small areas, depending on the

5 physical-geographical layout of the region. In the context of global warming, it follows that it is important to have an adequate monitoring network for these rain events, which are highly concentrated in space and time.

This research analyzes the meteo-hydrological features of August $27^{th}$ and $28^{th}$ 2023 event that occurred in Genoa, just 4 days after the record maximum air temperature recorded: between 19UTC and 02UTC nearly 400 mm of rainfall was recorded in the eastern sector of Genoa's historic center, with significant ground effects such as flooding and overflowing in

pressurized culverted waterways. Rainfall observations and estimates were taken using both official or "authoritative" networks (rain gauges and meteorological radar) as well as rain gauge networks inspired by citizen science principles.

The combined analysis of the observations by authoritative and "citizen science" networks highlights, for the analyzed event, a spatial variability of the precipitation field for the hourly and sub-hourly duration, which cannot be captured by the current spatial density of the authoritative measurement stations (which it is also among the highest in Italy). Monthly total rainfalls

and maximum intensity and short duration annual maxima time series recorded by the authoritative rain gauge network of the Genoa area are then analyzed. Results show that at distances even less than 2 km the variations in average rainfall depth cumulated over sub-hourly duration are very significant. Thus, extreme weather monitoring activity is confirmed as one of the most important aspects in terms of flood prevention and protection in urban areas. The integration between authoritative and citizen science networks can prove to be a valid contribution for monitoring this type of extreme events.





## 1  Introduction

The study of the impact of climate change on heavy precipitation phenomena at the global scale is an important field of research aiming at assessing possible changes in frequency, intensity, and in related spatio-temporal scales. Seneviratne et al. (2012) and Raymond et al. (2020), based on observational data, concluded that it is likely that the number of heavy precipitation events over land had increased in more regions than it had decreased, though there are wide regional and seasonal variations,

and indeed trends in many locations are not statistically significant. Du et al. (2019) and Dunn et al. (2020) showed that the annual average of the maximum rainfall amount in a day (Rx1day) has significantly increased since the mid-20$^{th}$ century over land. Furthermore, the globally averaged annual fraction of precipitation from days in the top 5% (R95pTOT) has also significantly increased (Dunn et al. , 2020). Sun et al. (2021), using high-quality station data up to 2018, studied changes in annual maxima of 1-day (Rx1day) and 5-day (Rx5day) precipitation accumulation at different spatial scales and they found

that the addition of the decade (2009-2018) of observational data shows a more robust increase in Rx1day over the global land region. When moving to the sub-daily rainfall, it is well known that the available data records are certainly shorter than what is requested for a robust quantification of past changes (Li et al. , 2019). Still, there are studies in regions of almost all continents that generally indicate intensification of sub-daily extreme precipitation such as in South Africa (Roy and Rouault , 2013), in Australia (Guerreiro et al. , 2018), in India (Ali and Mishra , 2018), and in eastern China (Xiao et al. , 2016). Furthermore,

Arnone et al. (2013) and Libertino et al. (2019) identified similar trends also in some regions of Italy.

This body of studies suggests the importance of an adequate monitoring capability to capture localized extreme rainfall events in a warming climate. Beside already consolidated remote sensing-based rainfall estimates, taking advantage of the combination of authoritative in situ weather stations with opportunistic networks inspired by citizen science principles (Bedrina et al. , 2013; Starkey et al. , 2017; Tipaldo and Allamano , 2017) is becoming more and more important.

Due to the limited extension of the urban catchment areas, with a high density of buildings and largely impervious surfaces, rapidly evolving pluvial floodings are typically experienced in cities resulting from the inefficiency of the urban drainage system in terms of the hydraulic failure of the storm water pipes and/or the insufficient capacity of the storm drain inlets. In the Mediterranean region, this is accompanied by a rainfall regime characterized by short-duration and high-intensity events, which typically show a quite limited spatial extension and very rapid evolution, therefore hard to be captured by the traditional

monitoring networks. Urban areas are also affected by riverine floods, from the overtopping of small streams (often covered and/or with narrowing sections) in orographically complex and steep areas. Since the storm water drainage in the urban texture is structured in a great number of small-size catchments, usually smaller than the typical spacing of rain gauge stations, urban catchments are often ungauged. Depth–Duration–Frequency (DDF) relationships derived from a rain gauge positioned in a conterminous urban catchment could estimate a return period associated to a specific rainfall depth that would affect

the design of the pipe size or of potential structural adjustments of the drainage network. The temporal resolution of rainfall measurements also plays a key role in the study of flood events: the faster the hydrological response of the basin, the finer is the required resolution of the supporting data. The World Meteorological Organization (WMO) developed the Observing Systems Capability Analysis and Review (OSCAR) tool (https://space.oscar.wmo.int/) that contains quantitative user-defined





requirements for observation of physical variables related to weather, water, climate, and other application areas. About the

55 variable "precipitation intensity" and the application area "Nowcasting/Very Short-Range Forecasting", that can be considered relevant for small catchments, OSCAR reports that the highest performance level (indicated with the term "goal") can be reached using measurements at a spatial resolution up to 1 km and a temporal resolution of 5 minutes.

Along these lines the characteristic spatial scales of the events that produce intense rainfall in the Ligurian area, compared with the observational capacities of the authoritative networks, are analyzed in this work and the example of the extreme rainfall

event occurred on August $27^{th}$ and $28^{th}$, 2023 in Genoa city centre is presented. Section 2 describes the study region, while Section 3 analyses the Genoa area rainfall regime. In Section 5 the characteristics of flood events in the historical centre of the study area are described. Section 4 describes the synoptic and mesoscale analysis during the event of August $27^{th}$ and $28^{th}$, 2023 and the observed impact at the ground . Section 6 reports the conclusions of the study.

## 2   The study region: Genoa Living Lab

Genoa is an ancient pre-roman city in which urban development was concentrated in the twentieth century (Faccini et al. , 2015, 2016). Genoa is the capital of the Italian region of Liguria and the sixth-largest city in Italy. In 2015, about 600,000 people lived within the city's administrative limits. As of the 2011 Italian census, the Province of Genoa, which in 2015 became the Genoa Metropolitan Area (GMA), had about 850000 resident persons.

The GMA is historically affected by flash floods and severe sea-storms for the meteorological conditions due to the "Genoa

Low" (also known as 'Ligurian Gulf Depression') and the city's geographical layout, which is characterised by a narrow coast bounded by steep mountains.

The 'Genoa Low' is a cyclone that forms or intensifies from a pre-existing cyclone to the south of the Alps with an orographic effect (Jansa et al. , 2014) over the Gulf of Genoa, Ligurian Sea. This depression is linked to the arrival of the Atlantic perturbations behind the Alps and is formed on the Gulf primarily in the autumn–winter and spring periods. This circulation is

75 responsible for the large amounts of rainfall distributed over the overall region surrounding the Ligurian Sea (Sacchini et al. , 2012).

The area of Genoa is characterized by a complex morphology due to the Alps–Apennines system, which hosts relief including peaks between 1000 and 2000 m above sea level and rapidly descends towards the Ligurian Sea. The resulting hydrographic network is made by numerous steep and short watercourses that can attain a concentration time between 3-4 hours down to less

than 1 hour. Therefore, the morphology of the Ligurian Gulf and the orographic barrier contributes to rainfall events that may be very intense, especially at the end of the summer or autumn, when Atlantic perturbations may be blocked by the European continental anticyclone. The convergence between the air mass stationed over the warm Mediterranean basin and the colder air masses moving from the Po basin triggers the development of convective systems very intense and localized, known as back-building mesoscale convective systems (Fiori et al. , 2014; Rebora et al. , 2013). These localized convective systems have

historically affected different locations over the Ligurian Gulf (Portofino 1915, Cinqueterre 2011, Genoa city 2011 and 2014),



causing flash floods arising from rainfall intensities of about 500 mm/6h or 180 mm/1h (Fiori et al. , 2014; Parodi et al. , 2017; Faccini et al. , 2021).

While the city of Genoa is internationally known for its recurrent floods, mainly related to the Bisagno River, high levels of river-related risk are also present in the historical city center. The watercourses of the historical city have been subject to relevant anthropic modifications since the Middle Ages; today the hydrographical network appears almost entirely artificial, flowing under the streets and buildings of the historical centre (Figure 1). The morphological amphitheatre, where the historical city of Genoa has developed over the centuries, represents as whole a composite catchment area that includes seven small and steep catchments that today are barely identifiable. The three catchments located in the eastern sector of the historical centre are the ones of greatest interest because they include the Medieval city and are most frequently affected by flooding phenomena. Besides, in a straight distance of less than 2 km from the coastline the catchments' heads reach altitudes between 170 m a.m.s.l. and 360 m a.m.s.l. The area of the Torbido stream catchment (Figure 1, catchment 1) covers about 1.17 km$^2$ and consists of two branches: a western and an eastern one that join underneath one of the major squares of the historical city. Then, the watercourse continues under the main streets of the area and flows at the docks. In 1519 the watercourse was still uncovered while today the riverbed is entirely culverted, with a rectangular cross-section and an estimated flood discharge for T=200 years of 55 m$^3$/s. The Sant'Anna stream catchment area (Figure 1, catchment 2), which covers 0.72 km$^2$, is also composite, as it results from the junction of two small branches. After the junction it flows under the UNESCO area of the historical centre. The lower part of the catchment, partly due to inadequate hydraulic section, is characterized by flood hazard: the estimated maximum flow rate for T=200 years is 44 m$^3$/s (Autorità di Bacino , 2023).

The Carbonara Stream basin (Figure 1, catchment 3) is articulated too, as its hydrographical network includes the San Gerolamo Stream on the east, formerly independent and then diverted into the Carbonara stream itself (1.10 km$^2$). In the 13$^{th}$ century, the riverbed was diverted at the heart of the historical city centre and the diversion made this creek a tributary of the Carbonara one. The Carbonara catchment presents two small valleys in its upper part. Their main channels join in a collector that flows into the sea in the historical city harbour. In its middle stretch the riverbed shows several hydraulic works, such as weirs, waterfalls, settling chambers, spillways and anthropogenic secondary branches. In 1336 the Carbonara Stream was uncovered, while today the entire riverbed is culverted. The final stretch has a reduced hydraulic section due to foundation structures that partially obstruct the conduit; the flood discharge for T=200 years is estimated to reach 68 m$^3$/s. The construction of these culverts over time and the modifications they have undergone over the centuries up to very recent times due to progressive urbanisation (at present day 75%) led to a reduction in the riverbed cross-section, which can lead to possible overflows of water under pressure and consequent floods. Figure 2 shows the anthropogenic modification degree at the catchment scale, while the perspective view in Figure 1 allows to appreciate the buildings saturation of the historical city centre and the culverted streams path.





## 3 The Genoa area rainfall regime

The spatial variability of intense rainfall events in the Genoa city center is significant, even across a limited portion of the territory (Cauteruccio et al. , 2023). To corroborate this statement, five rainfall time series of at least 12 years of high-resolution records, from rain gauges positioned in the town of Genoa, are analysed and compared in this section. Accurate rainfall intensity (RI) measurements (Lanza and Cauteruccio , 2022) and their spatial and temporal resolution play a key role in the modelling of flood events in urban areas (Lin et al. , 2022; Bruni et al. , 2015). The large spatial variation of rainfall intensity, especially for short duration events, is shown below to arise from the analysis of rainfall statistics obtained using rainfall records from conterminous rain gauge stations in the town of Genoa. A reference station was selected based on the high accuracy and fine resolution of the available data. Four further rain gauge stations were selected within a radius of 1.4 to 7.9 km from the reference station. Monthly rainfall totals and extreme value statistics of short-duration/high-intensity events were compared to show that, even within such a short distance the variation of the expected maximum rainfall is significant, as shown in the following paragraphs. As a reference data set, about thirty years of rainfall intensity measurements (from January $1^{st}$, 1988, to December $31^{st}$, 2021) at one minute resolution are available from the tipping-bucket rain gauge (TBRG) of the meteorological station of the department of Civil, Chemical and Environmental Engineering (DICCA) at the University of Genova (manufactured by CAE S.p.A – Italy). These are available as raw and corrected data, after suitable calibration curves are applied, as derived at the laboratory of the Measurement Lead Centre on Precipitation Intensity of the World Meteorological Organization (WMO). The applied calibration and adjustment are compliant with the European norm EN 17277/2019 (CEN, , 2019). The typical instrumental bias of TBRGs (see e.g., Cauteruccio et al.  (2021)) affects the high rainfall intensity measurements (RI > 150 mm/h) with underestimation larger than 5% while, after appropriate adjustment the measurement bias is contained between $\pm 3\%$ within the entire measurement range. The three further rainfall series analysed in this work were obtained from the authoritative monitoring network of the regional Agency for Environmental Protection of the Liguria region (ARPAL) at the temporal resolution of five minutes. Twelve years of high-resolution data (2011 to 2022) were available from the stations named Centro Funzionale (CF), Sant'Ilario (SI) and Castellaccio (CA), located within a short distance from the reference station (see Table 1 for the location and distances of the investigated stations). Finally, hourly rainfall data for the same period were also obtained from a rain gauge station named Genova University (GU), owned by the same University, located within the historic centre of the town. The name of each station with the associated acronym and the characteristics of the available rainfall measurements are listed in Table 1. The DDF curves were derived using the 30-year time series of corrected one-minute RI measurements from the DICCA reference rain gauge. Due to the relatively short duration of the time series available from the nearby stations, only the DDF curves associated with return periods T = 1.5, 2.32 and 5 years were used in the analysis. Moreover, these are typical values of the return period assumed to design urban drainage networks, beyond which the failure of the drainage system is accepted. Note that a return period of 2.32 years was chosen because it is associated with the mean value of the sample of annual maxima calculated per each event duration (d) under the hypothesis of an underlying Gumbel distribution of rainfall extremes (Gumbel , 1941).





150 The year-to-year variability of monthly precipitation for twelve years of records at the four rain gauge stations studied is shown in Figure 3. Boxes and whiskers cover the central 50% and 80% of the data set, respectively. The mean and median values are indicated in each box by the thick and thin horizontal lines, respectively. The mean values, the extremes and the year-to-year variability observed at SI are systematically higher than at the other stations. In dry summer months, the relative difference between the mean precipitation at two conterminous stations is limited, but in the wet months (autumn and spring),

155 when intense precipitation is mostly expected, this difference is enhanced, notwithstanding the short distance among them. In autumn, in particular, the mean value and the year-to-year variability of the annual rainfall at GU is systematically lower than at the other stations. Differences in the precipitation regime at the considered stations are also evident for short-duration high-intensity events, as shown in Figure 4 based on the depth duration frequency (DDF) curves calculated at the reference nearby station (DICCA). Short duration events, for 5, 10, 15, 30, 45 and 60 minutes were considered in the analysis, therefore

160 excluding the GU station due to the coarse time resolution (1 hour) of the available data set.

 The mean value of the annual maxima at each station and for each duration can be compared with the reference DDF curve at T = 2.32 years. The percentage difference $e$ [%] between such mean values at each of the investigated stations and at the reference station, was therefore calculated (see Table 2). The comparison shows that short-duration high-intensity events at the Arpal CF station are up to 37% lower than the typical events at the reference station for any duration. Meanwhile, the statistics

165 at the Arpal SI station are close to the reference station for events of short duration (between 10 and 30 minutes), while longer events are less intense, at least over the investigated period. The Arpal CA station shows an intermediate behaviour. Results show that, for each duration, the mean values of the annual maxima at the Arpal CF gauge are characterized, in the observed period, by the same return period T = 1.5 years, while the Arpal SI station exhibits the largest annual maxima for short duration events (up to 15 minutes) with T = 2.32 years, then the return period decreases with increasing the event duration, and reaches

170 1.5 years at d = 60 minutes. For the Arpal CA station, the mean annual maxima range from above to slightly below T = 1.5 years with increasing the event duration. Note that, for short duration events (shorter than 15 minutes), the Arpal SI station is the closest to the reference station in terms of the return period of its annual maxima, though being the farthest in geographical terms. The dispersion of annual maxima around the mean value for this station however reaches the largest return period investigated (T= 5 years) with the limits of the interval covering 80% of the sample, which is not the case for any other station

175 in the range of short duration events.

 A large variability is therefore observed between the investigated stations especially for short duration, which are typically associated with critical conditions in urban drainage systems. The variability of extreme value statistics between the investigated rain gauge stations over the considered period reflects the spatial variability of short-duration high-intensity events, even at the limited spatial scale of urban catchments. In light of this analysis and for the high spatio-temporal variability of severe

180 events over the Genoa city center area, a specific case study recently occurred on August $27^{th} - 28^{th}$, 2023 is discussed below, where the historical centre and the San Fruttuoso district of the town of Genoa suffered from intense pluvial flood events that were not observed in nearby districts of the town.





## 4    Flood events in the Genoa region

As explained in the introduction, there is evidence that climate change can also cause an intensification of short-lived and
highly intense events in Italy and the Mediterranean area (Arnone et al. , 2013; Libertino et al. , 2019).

The analysis carried out in the previous section shows how, in a highly urbanized area and with a drainage network now
completely artificialised, a structure common to many historic cities of the Mediterranean, events of this type present charac-
teristic spatial scales such as to make them not completely observable from standard authoritative measurement networks. A
better knowledge of the spatial scale and probabilities of occurrence of short-duration heavy rainfall is fundamental for the
correct design of risk management measures in urban areas and the role of observations provided by networks managed by
citizen scientists can prove fundamental.

In historic centres like the Genoa one, the channelized drainage networks have often under-dimensioned hydraulic sections
as they have been designed with antiquated methodologies and a lack of observational data. Furthermore, they have been
built in times when the geomorphoclimatic context was different: impervious surface areas have changed over time and the
precipitation regime has probably changed over the centuries.

As mentioned before, Genoa is a Mediterranean city in which climate change is evidenced by an increase in average air tem-
perature and a change in rainfall patterns. Recent years have experienced an increase in flooding episodes in the metropolitan
city of Genoa, concentrated mainly in the autumn period and often characterised by exceptional rainfall values with records
even at the international level: 181 mm/1h were recorded in November 2011, while in October 2021 378 mm/3h, 496 mm/6h,
741 mm/12h, and 884 mm/24h were measured (not far from the Italian historical record of 948 mm registered in October
1970 in Genoa again). Flood hazard in the historical city, which has been particularly frequent in the past 30 years, is mainly
associated with pluvial flooding because of the inefficiency of the urban drainage system.

In situations such as those of Genoa's historical centre, with such a high anthropogenic impact on the natural morpho-
genetic system, it is tough to design risk mitigation actions: it is not possible to operate with structural interventions aimed
at increasing the hydraulic section due to urbanization, nor it is possible to reduce the impervious area significantly. Possible
solutions include the so-called "green infrastructures" (e.g. porous materials for pavements, green roofs) or old cisterns con-
nected to buildings as temporary floodwater storage basins to attenuate the peak discharges, which is usually very rapid given
the physical-geographical layout of urban areas such as the Genoa's historical centre.

## 5    Analysis of the event of August $27^{th}$ and $28^{th}$, 2023

### 5.1    Ground effects

Local newspapers and the event report of ARPAL (ARPAL , 2023) describe the impact of the investigated event on the involved
catchments. The event produced severe hydrological impacts in the Genoa city center including the flooding of the Genova
Piazza Principe railway station around 23UTC, even if the railway traffic was not interrupted. The Genoa subway services and
the Sant'Anna and Zecca–Righi funiculars services were stopped for a few hours during the night. Dozens of interventions




by the firefighters due to flooding of cellars and garages were reported in the districts of Oregina and Castelletto, but also in
the Foce district and in the historic center. There were also many calls to rescue people stuck in the water near underpasses.
Several cars were flooded. There were also many interventions due to the collapse of retaining walls and trees that fell due to
strong winds. Some small natural streams (among them the Lagaccio river, see Figure 15) reached the overflow warning level.
In Figure 16 some pictures that captured the post-event evidences are reported.

### 5.2 Synoptic and mesoscale analysis

On August $25^{th}$, a North Atlantic disturbance began to descend towards the northwestern Alpine arc, weakening the northern-
most part of the high-pressure promontory of North African origin present on the central Mediterranean. This synoptic scenario
caused a rotation of the currents from the southern quadrants. This rotation favoured the entry of a Warm Conveyor Belt char-
acterized by moist air moving northward from the Ligurian Gulf. The warm and humid air continued to flow over the region

for the following days and caused high instability further north in the Alpine foothill areas and the Po Valley (Figure 5). From
the evening of August $26^{th}$, the Atlantic trough extended further south, reaching the latitudes of the Iberian Peninsula and, on
the following day , a Low structured at all altitudes isolated itself from the trough, moving progressively from Spain towards
the Ligurian gulf, where it arrived on the night between August $27^{th}$ and $28^{th}$ (Figure 6). The influx of humid air from the east
began even more pronounced onto the Po Valley and a pressure gradient developed on the mountains favouring north winds on

the western part of the Genoese area, and east winds on the eastern part of the Ligurian territory. The subsequent convergence
over the sea caused by this situation, together with the influx of humid air from the sea, exacerbated the atmospheric instability
over the sea during the night hours, resulting in the formation of cumulus clouds which caused thunderstorms moving from the
Ligurian Sea towards the coast, with showers developing on the historic centre of Genoa.

  From the analysis of the possible air column starting from the soundings of Novara Cameri and Cuneo Levaldigi (Figure

7), it appears that in the afternoon of August $27^{th}$, while in the southern part of the Ligurian gulf the instability indices do not
highlight a particular situation (Payerne), on the north-west of the peninsula there was high energy available for convection
(CAPE 1800 J Cuneo, 2700 J Novara), with notable instability indices (Lifted index -4 Cuneo, -7 Novara, wind shear above
20 m/s). These values of instability indices suggest the possible formation of organized thunderstorms with a high risk of very
intense rain.

During the following day on August $28^{th}$, the minimum pressure moved eastwards, giving way to drier northerly currents
which resulted in a dry Tramontana wind with consequent improvement in the weather.

### 5.3 Rainfall phenomena

After a pre-frontal phase of thunderstorm nature between the night on Saturday, August $26^{th}$, and the early hours of Sunday,
August $27^{th}$, the most intense precipitation effects were observed on Sunday night, between $27^{th}$ and $28^{th}$ August, especially

in the Genoese area. Around 09UTC eastern Liguria started to be interested by thunderstorm activity with the main convective
cell in front of the Portofino area and associated rainfall intensity around 30-40 mm/h (Figure 8). One hour later, 10UTC, the
convective cells while persisting on the Portofino area started to affect also the Genoa city area with overall rainfall intensi-



ties peaking approximately up to 30-40 mm/h. The convective line originally present on the Portofino area gradually moved eastward affecting the areas of Chiavari, Lavagna and Sestri Levante (11UTC), and finally leaving eastern Liguria (12UTC).

Convective phenomena over the Genoa Gulf area dissipated during the afternoon until 17UTC when a new convective system, associated with intense lightning activity, moving from the French area intensified over the Ligurian sea about 100 km far apart the Genoa city area with rainfall intensity above 100 mm/h (Figure 9). Newly developed severe convection activity started to affect the Genoa city area around 17UTC and it persisted, in this phase, until early hours of August $28^{th}$.

    When focusing on the ground effects, different weather stations, belonging either to authoritative or citizen science networks

can be considered for the Genoa city centre area (Figure 10): 3 stations managed by Arpal (AR-1, AR-2 and AR3), as well as several citizen science stations (CS hereafter) provided by Acronet network (Fedi et al. , 2013), Meteonetwork (Giazzi et al. , 2022) and Wunderground (Gharesifard et al. , 2017) providing a high-density coverage of the historical city center of Genoa (stations CS-0 to CS-5).

    Acronetwork is composed by weather stations inspired by the ACRONET paradigm, a model of design, realization, diffusion

and installation in the field of environmental monitoring and control systems released according to an open hardware mode. A series of sensors or third-party devices can be connected to the board at the same time (with different technologies); each singular combination is a different configuration. The ACRONET station, considered in this work and located in the Porto Antico area (CS-0), involves the following third parties sensors: the LAMBRECHT 15189 precipitation sensor (resolution 0.1 mm, accuracy ±2% with intensity correction), the thermohygrometer SIAP+MICROS TTEPRH (resolution 0.03°C, accuracy

±0.1°C for temperature; resolution 0.5%, accuracy ±2% for relative humidity), and anemometer SIAP+MICROS WINSON (resolution 0.01 m/s, accuracy ±2.5%). Four stations (CS-1, CS-2, CS-3 and CS-4) contributing to Meteonetwork and Wunderground are located in the Genoa City centre and they correspond to the DAVIS Wireless Vantage Pro2 model: precipitation sensor (resolution 0.2 mm, declared accuracy ±3%), thermohygrometer (resolution 0.1°C, accuracy ±0.1°C for temperature; resolution 1%, accuracy ±2% for relative humidity), and anemometer (resolution 0.4 m/s, ±5%). The authoritative weather

station of ARPAL, located in Castellaccio area (AR-2, Figure 10) recorded a cumulated rainfall depth between 17UTC and 21UTC of about 120 mm (Figure 11). Subsequently convection phenomena reinforced again between 23UTC on $27^{th}$ August and 00UTC on $28^{th}$ August, leading to a total rain depth of about 200 mm in 6-7 hours (Figure 11). Other ARPAL stations (AR-1 and AR-3) recorded over the same time window about respectively 130 and 170 mm (Figure 11). However, beside ARPAL station(s), there also other observational networks near the Genoa city centre, corresponding to citizen science philosophy.

In fact, the two stations CS-0 and CS-5, about 600 m apart and located in the harbour "Porto Antico di Genova" area (Figure 10), recorded a cumulated precipitation of 381,3 mm and 380 mm respectively, almost double the ARPAL AR-2 station more than two km away (Figure 11). The latest has precipitation values more similar to the CS-1 station which is about 700 m away from it. More in depth, looking at the graph in Figure 11 showing the cumulated precipitation of all the stations, we can group them into three main sets, each one representative of the main showers associated with the event. The first one located in a

radius of less than 800 m from the centre of the main shower (CS-0 and CS-5, the Porto Antico harbour) coloured with different shades of yellow, another one eastward of it and in a radius of less than 2 km and coloured with different shades of blue (CS-2, CS-3 and CS-4), and the last one further away in a radius beyond 2 km towards the west and east of the area, as highlighted





with different shades of pink (ARPAL AR-1 and AR-3). From this punctual analysis and from the statistical analysis done in the previous paragraph a question arises. What would happen if data from citizen science stations were included in the process

of acquisition and production of the radar derived rainfall maps? Would the phenomenon be described more precisely from the point of view of the spatial distribution of precipitation? To do this we produced 12 hours cumulated precipitation maps from 18UTC of the August $27^{th}$ to 06UTC of August $28^{th}$, 2023 in two different configurations (Figure 12):

- merging only radar data and the official weather stations (Bruno et al. , 2021);

- adding also citizen science.

It is evident from the dataset analysed in this work that a higher density of rain gauge stations is required to capture the spatio-temporal dynamics of intense rainfall events that are prone to produce flooding in cities and critical conditions in stormwater drainage systems. Authoritative monitoring networks are often too coarse to provide sufficient coverage. Looking at the radar images in more detail, it is possible to gain a deeper understanding of the convective phenomenon affecting from the evening (19UTC) of August $27^{th}$ the Genoa historical centre. In fact, by making a cross-section of the echoes of the radar

that insists on the area of the Genoa ancient port, one notices how the column of air above is affected by extremely localized thunderstorms over a thickness of up to 8000 m (19UTC, Figure 13).

Taking into consideration the same cross-section in the period between 23UTC and 00UTC, we note how the convective structure is getting more organised, with a movement towards the East affecting a vertical portion up to above 10000 m. From the vertical section at 23:55UTC one can also appreciate the hook shape of the top of the convective cloud, typical of

300 meteorological structures with moist deep convection, causing high vertical movement of the air up to the boundary layer (Figure 14).

The comparison among the cumulated rain recorded by the different networks shows how the convective nature of the phenomenon reflects on extremely diverse effects on the territory as well as an almost punctual localization of the most intensive showers. The extremely punctual localization of this event shows how important it is to couple different monitoring networks

if present in the area to describe the event more precisely.

## 6 Conclusions

This study analyzed the role of citizen scientist stations in the observation and analysis of the spatio-temporal characteristics of intense and short-lived precipitation events, potentially dangerous for inhabited centres, especially in areas with complex orography.

The case of the historic center of Genoa (Italy) was used as an example. An event recently occurred between the $27^{th}$ and the $28^{th}$ of August 2023 was analyzed from a meteorological point of view with the help of radar, satellite and ground observations.

The observations from the closest radio soundings clearly show a high possibility of the formation of intense convective structures, which then actually formed over the Ligurian Gulf. Observations from the authoritative network and estimates from
315 meteorological radar only partially capture the spatiotemporal structure and maximum intensity of the forecasts generated over the Genoa area, mainly due to the coarse spatial resolution of both measurement systems. The inclusion of observations from "citizen science" measuring stations shows how in this case the precipitation field shows a higher spatial variability with values of cumulative precipitation on the event extremely variable within 1-2 km. This demonstrates how both the meteorological radar, which has a spatial resolution usually of the order of km, and the authoritative ground measurement networks, which,

in the best cases, in Italy, have densities of around 0.03 stations per km$^2$ (Pelosi et al. , 2020), are not suitable for adequately describing the precipitation field.

A further demonstration of this inadequacy not referring to a single event is obtained from a statistical analysis of the precipitation maxima for sub-hourly duration carried out on historical time-series recorded by authoritative stations in the Genoa area. In terms of the average depth of maximum annual precipitation for sub-hourly duration, differences generally

exceeding 20% are estimated even over distances of 2-4 km.

The inadequate observation of short-term intense rainfall has a dual impact: it can cause an underestimation of the design hyetographs for urban infrastructures or green solutions where the historical urban stratification does not allow interventions to modify the drainage networks and can limit the effectiveness of efforts to improve the reliability of high-resolution modeling of convective systems by lacking a reliable ground truth against which to test model performance.

In the case presented here, the introduction of citizen science observations allowed a better understanding of the spatio-temporal structure of the phenomena that caused urban flooding in the study area. In the future, a more structured use of this information, together with observation validation procedures, can provide a fundamental contribution to risk prediction and management in densely urbanized areas such as the historic centers of many Mediterranean coastal cities.

*Author contributions.* AP, NL and MM co-conceived the research, draft the paper, contributed to the meteorological analysis and revised
the manuscript. AC conceived and performed the analysis of the study area rainfall regime, drafted the related section and revised the manuscript. GB, FF and GP contributed to the description of the study region and revised the manuscript. All authors have read and agreed to the published version of the manuscript.

*Competing interests.* The authors declare that they have no conflict of interest.

*Acknowledgements.* This study was carried out within the RETURN Extended Partnership and received funding from the European Union
Next-GenerationEU (National Recovery and Resilience Plan – NRRP, Mission 4, Component 2, Investment 1.3 – D.D. 1243 2/8/2022, PE0000005) as well as supported by the European Union's Horizon 2020 I-CHANGE project (https://cordis.europa.eu/project/id/101037193/it). We extend our sincere appreciation to ARPA Piemonte for their essential support in providing atmospheric soundings, which greatly enriched the scope of our research. Additionally, we would like to acknowledge ARPA Liguria for their valuable contribution in supplying radar cross



sections. The collaboration with both ARPA Piemonte and ARPA Liguria has been instrumental in advancing our understanding of the
atmospheric conditions crucial to this study.



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





| Station name (and acronym) | N | E | Elevation [m] A.M.S.L. | Distance from DICCA [km] | Time resolution | Available period |
|---|---|---|---|---|---|---|
| DICCA rain gauge (DICCA) | 44°24'0.1" | 8°57'48" | 45 | - | 1 min | 1988 - 2021 |
| Arpal Centro Funzionale (Arpal CF) | 44°24'1.26" | 8°56'45.276" | 30 | 1.4 | 5 min | 2011 - 2022 |
| Arpal Sant'Ilario (Arpal SI) | 44°23'2.4" | 9°3'38.376" | 174 | 7.9 | 5 min | 2011 - 2022 |
| Arpal Castellaccio Arpal CA) | 44°25'40.692" | 8°56'3.588" | 360 | 3.9 | 5 min | 2011 - 2022 |
| Genova Università (GU) | 44°24'55" | 8°55'38" | 58 | 3.3 | 1 h | 2011 - 2022 |

**Table 1.** Location of rain gauge stations used to highlight differences in intense rainfall statistics and characteristics of the available rainfall measurements.

| | d=5 minutes | d=10 minutes | d=15 minutes | d=30 minutes | d=45 minutes | d=60 minutes |
|---|---|---|---|---|---|---|
| e(CF-ref)/ref [%] | -37 | -26 | -22 | -20 | -25 | -25 |
| e(SI-ref)/ref [%] | -24 | -9 | -4 | -12 | -21 | -26 |
| e(CA-ref)/ref [%] | -26 | -15 | -14 | -20 | -25 | -28 |

**Table 2.** Percentage difference between each of the investigated stations and the reference station, for a 2.32-year return period (mean annual maxima).

425




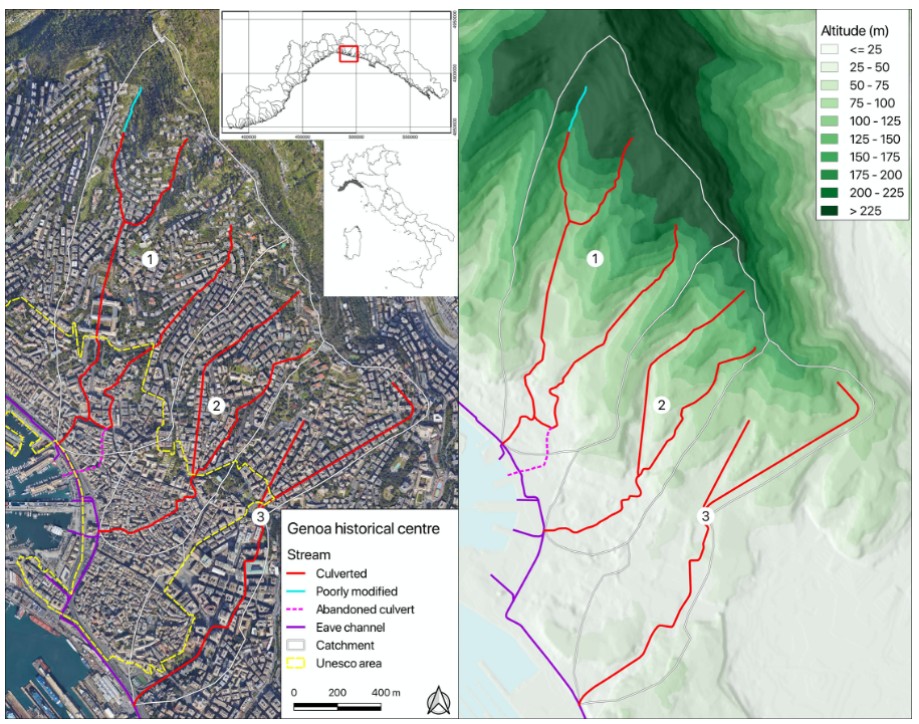

**Figure 1.** The Torbido stream catchment (1), Sant'Anna (2) and Carbonara (3) locations in the Genoa historical centre and related subterranean waterways (left panel, © Google Earth) and corresponding digital terrain model (right panel).

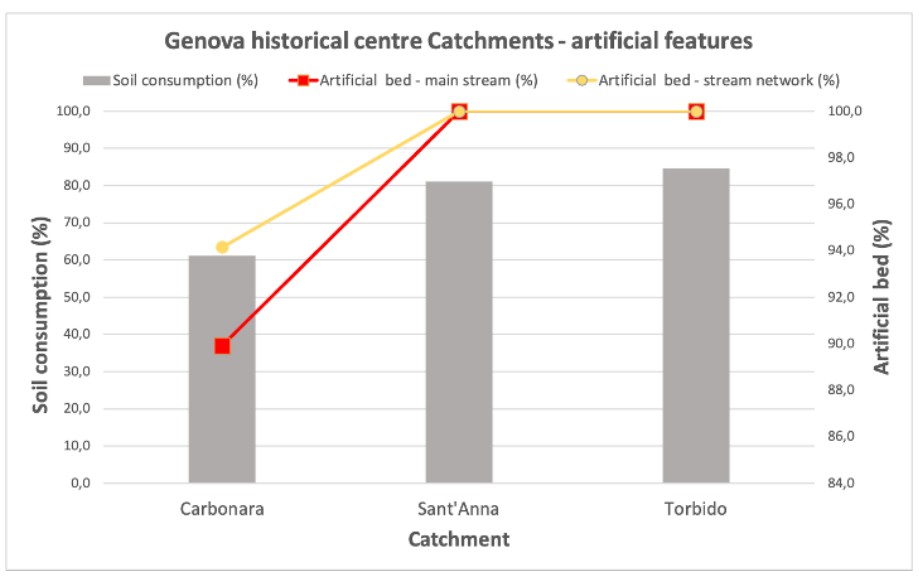

**Figure 2.** Identification of the main artificial features for the Carbonara, San'Anna and Torbido catchments.

Natural Hazards
and Earth System
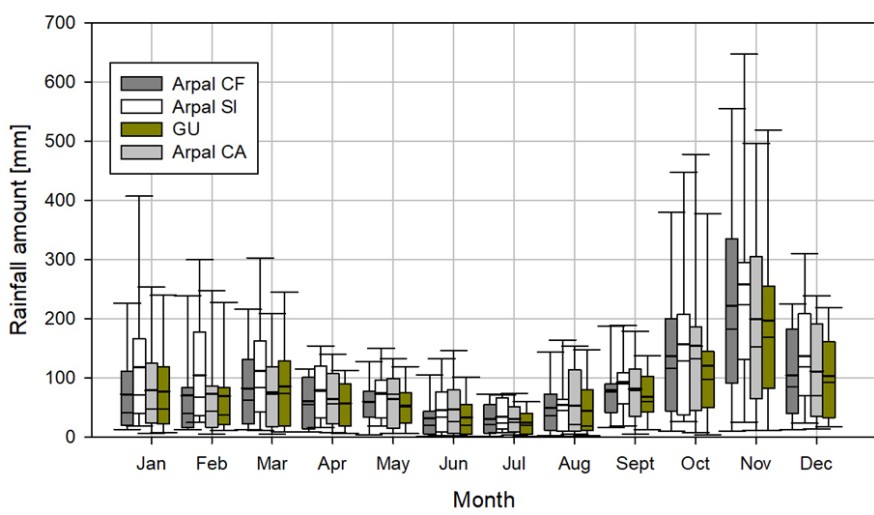

**Figure 3.** Year-to-year variability of monthly precipitation at the four rain gauge stations studied. Boxes and whiskers cover the central 50% and 80% of the data set, respectively. The mean and median values are indicated in each box by the thick and thin horizontal lines, respectively.

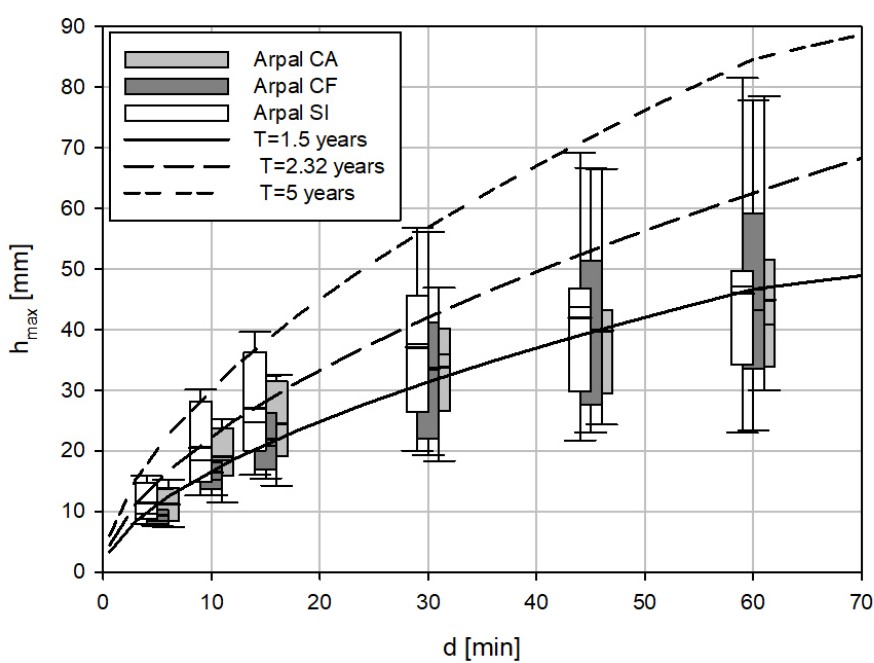

**Figure 4.** Annual maximum rainfall depth for each duration at the three Arpal rain gauge stations studied (boxplot), overlaid with the DDF curves derived at the DICCA reference station. Boxes and whiskers cover the central 50% and 80% of the data set, respectively. The mean and median values are indicated in each box by the thick and thin horizontal lines, respectively.

**Figure 5.** MSG (Meteosat Second Generation) infrared images (10.8MHz) at 22UTC (upper left), 23UTC (upper right) on $27^{th}$ August 2023 and at 00UTC (bottom left) and 01UTC (bottom right) on $28^{th}$ August.





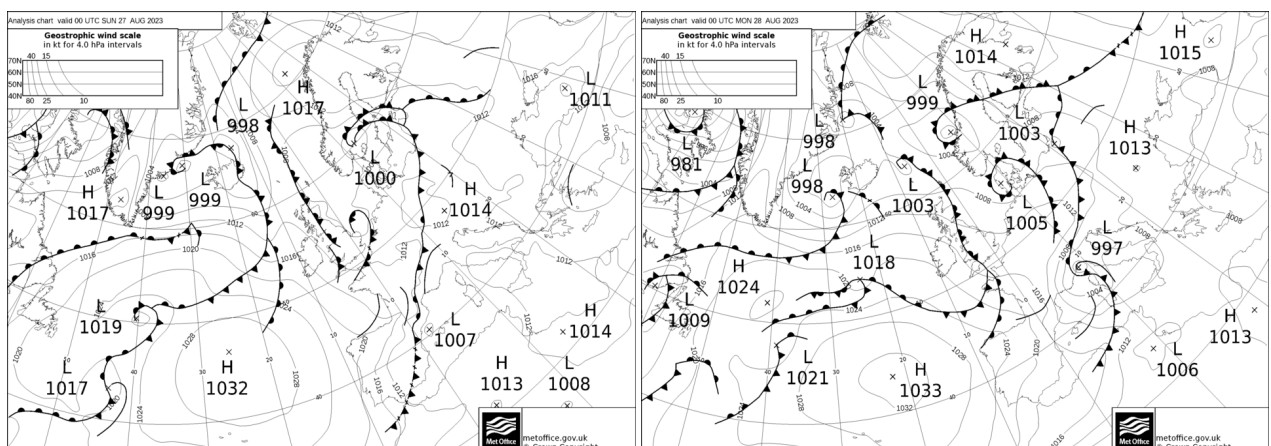

**Figure 6.** Analysis chart for 00UTC on $27^{th}$ August 2023 (left) and $28^{th}$ August 2023 (right).

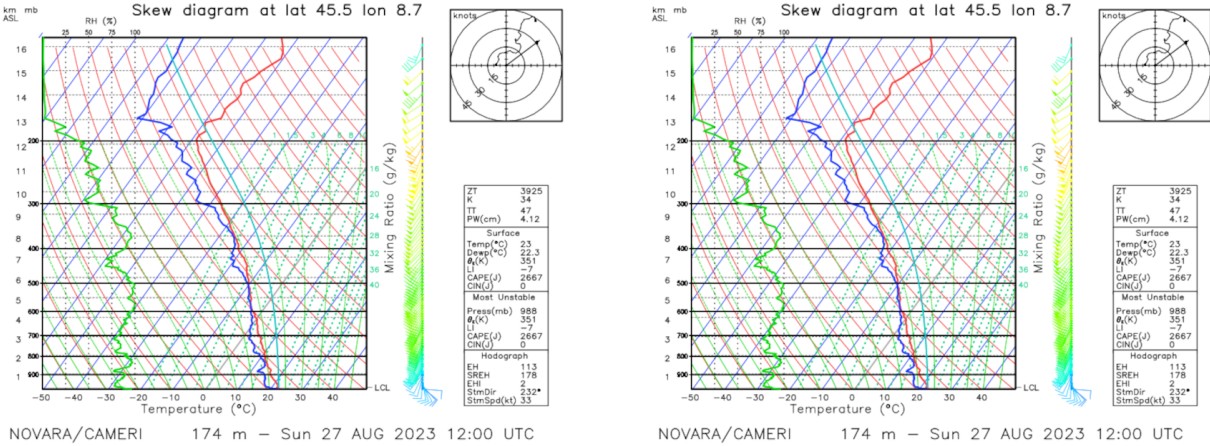

**Figure 7.** Skew-T diagram in Cuneo Levaldigi (left panel) and Novara-Cameri (right panel) on $27^{th}$ August 2023, 12UTC.

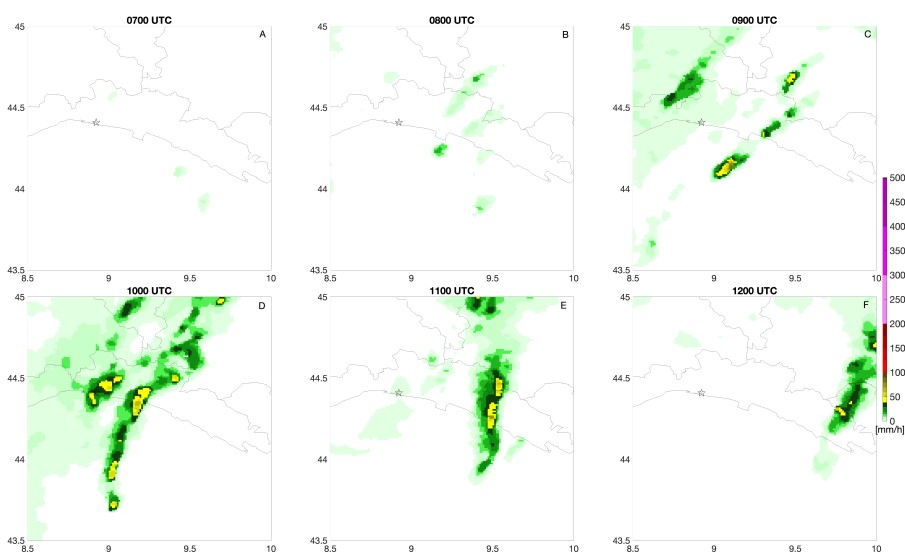

**Figure 8.** Radar derived rainfall intensity maps (7-12UTC, from top left to bottom right) on $27^{th}$ August 2023. The star marker shows the Genoa historical city centre position.

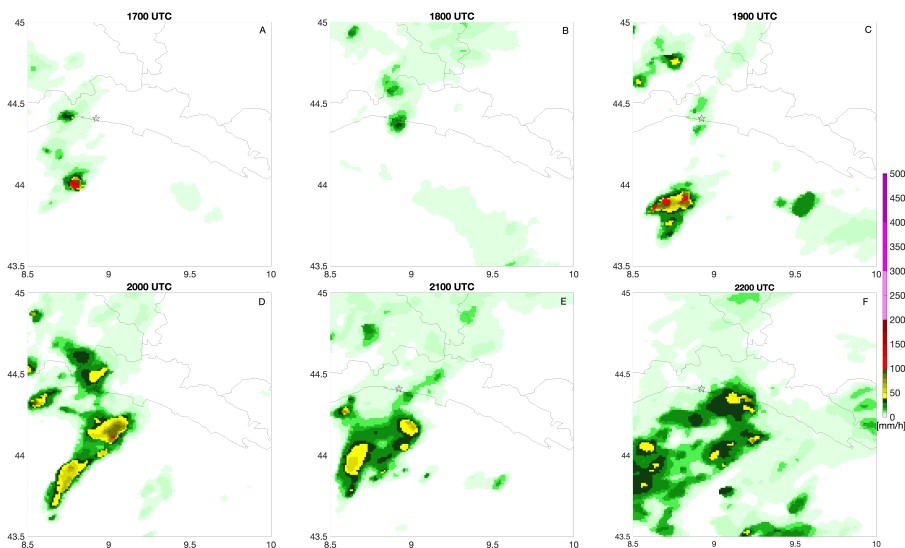

**Figure 9.** Radar derived rainfall intensity maps (7-12UTC, from top left to bottom right) on $27^{th}$ August 2023. The star marker shows the Genoa historical city centre position.


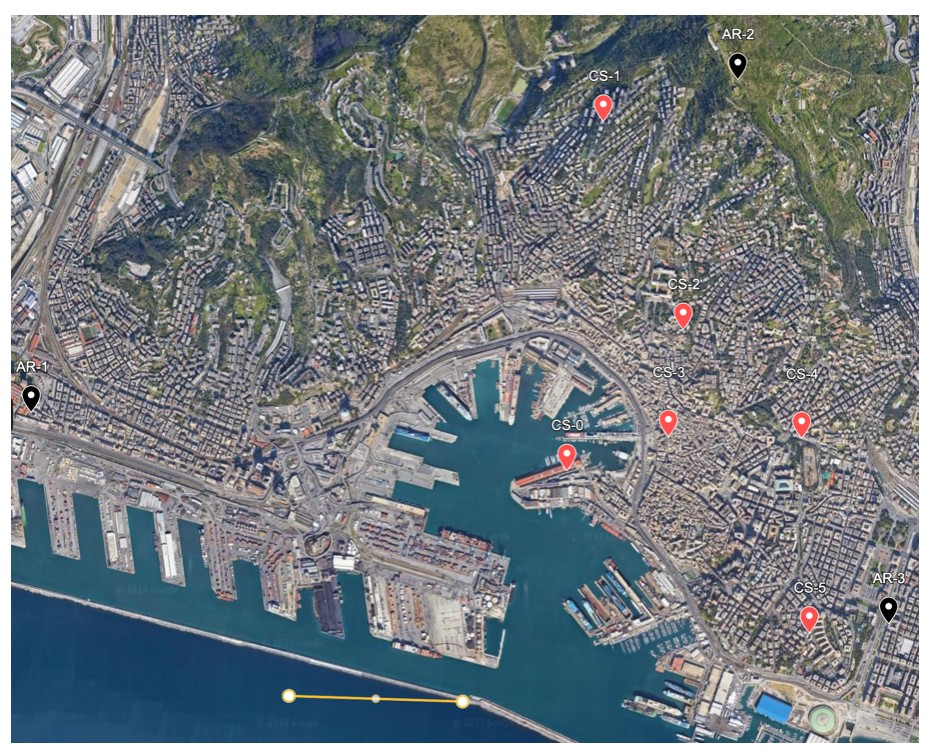

**Figure 10.** Rain gauge stations (authoritative and citizen science) in the Genoa City Center (© Google Earth).

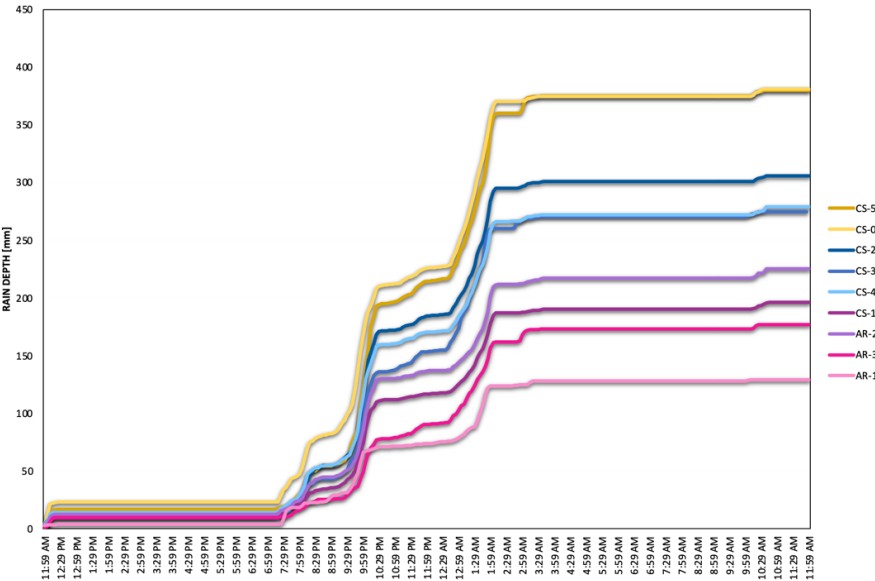

**Figure 11.** Rain depth recorded from authoritative and citizen science networks in the Genoa city center.


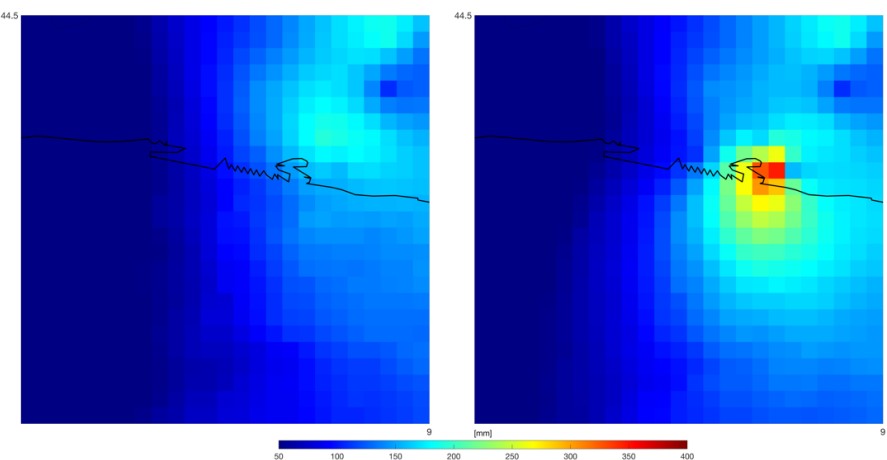

**Figure 12.** 12 hours rainfall depth (18UTC $27^{th}$ August – 06UTC $28^{th}$ August 2023, based on the merging of radar estimates and authoritative rain gauges (left panel) and adding also citizen science ones (right panel).

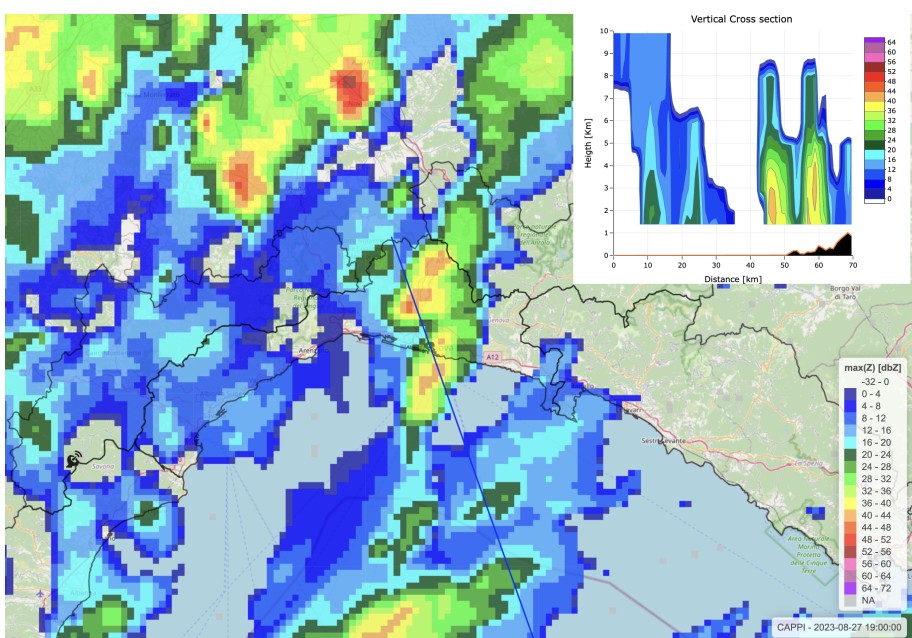

**Figure 13.** Radar reflectivity map (19UTC) on $27^{th}$ August 2023 and relative cross-section.

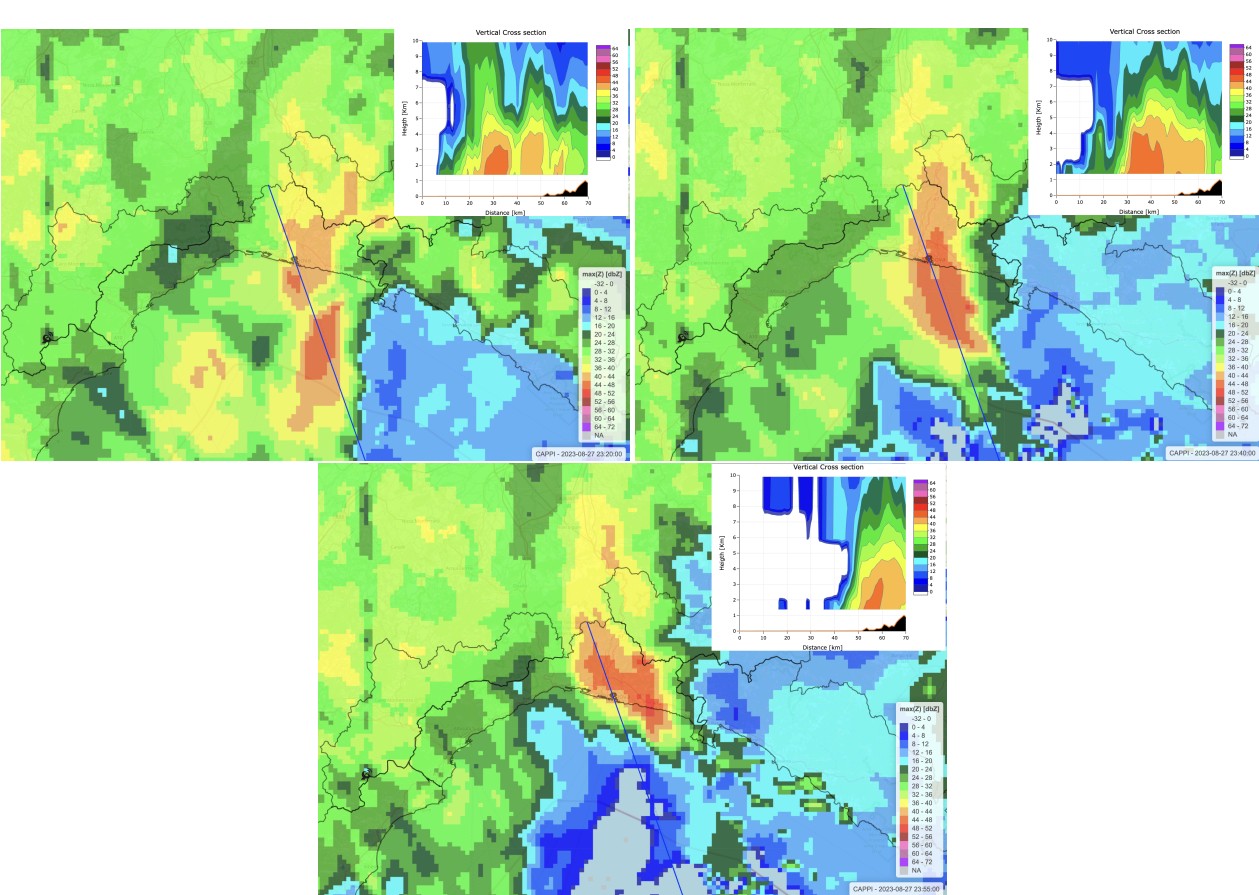

**Figure 14.** Radar reflectivity maps (23:20, 23:40 and 23:55UTC) on 27$^{th}$ August 2023 and relative cross-sections.



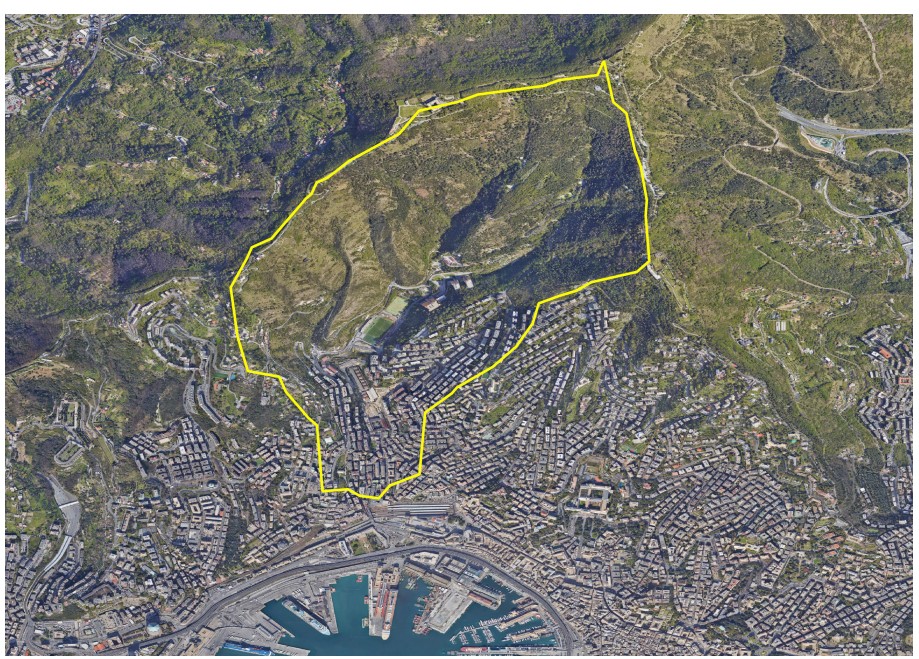

**Figure 15.** The Lagaccio creek catchment. The lower right part of the image includes the Genoa historic city centre (© Google Earth)

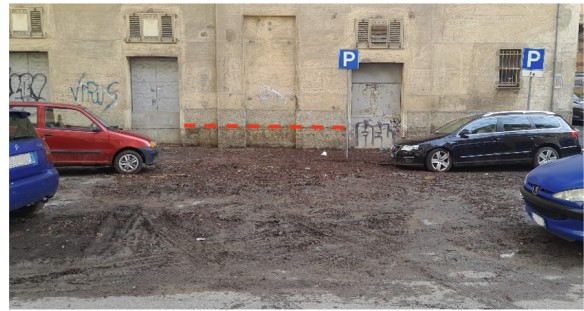 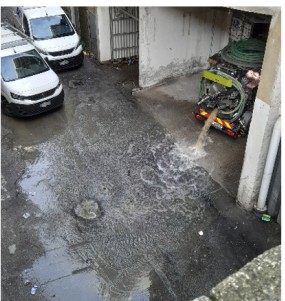

**Figure 16.** Pictures showing post-event traces of the ground effects of the event analysed. The red dashed line shows traces of the water level reached during the event