# Peer review of "The role of citizen science to assess the spatiotemporal pattern of rainfall events in urban areas: a case study in the city of Genoa, Italy"

_Natural Hazards and Earth System Sciences, 2024_

## Author Response (AR1)

This study analyzed the role of citizen scientist stations in an extreme rainfall event, which can be beneficial for promoting the utilization of citizen station data for precipitation research. However, there are several concerns that need to be addressed before considering this manuscript for publication.

Major Comment :

1. It is suggested to increase the **Data** This section focuses on station information, especially the accuracy of the rain gauges used. And the meteorological and remote sensing data source and access was not mentioned in this paper. The missing part can also be added into **Data** section.

   A data section was added (named Data Sources) to describe the characteristics of station data as well as meteorological and remote sensing data sources and access information.

   *Radar data source:*

   *Bruno, G., Pignone, F., Silvestro, F., Gabellani, S., Schiavi, F., Rebora, N., ... & Falzacappa, M. (2021). Performing hydrological monitoring at a national scale by exploiting rain-gauge and radar networks: the Italian case. Atmosphere, 12(6), 771.*

   *Satellite Meteosat Second Generation are available in the MyDewetra platform*

2. The station datasets are often considered as the true value. Thus, the assessment of station datasets is pretty difficult. This paper combined many meteorological and remote sensing datasets to evaluate the station datasets. The precise representation of spatial distribution is crucial here. However, the coverage ranges of many figures in the manuscript appear inconsistent, and there is a lack of information within the figures to indicate their coverage ranges. This is detrimental to the analysis of assessment results. I will provide specific details in minor comments.

   Figures have been modified to have more focused coverage ranges and to include the missing information.

Minor Comment:

1. Shen (2015) also investigated that adding more station as fusion data can enhance the accuracy of the precipitation data. Is the necessity of a denser network of station data for urban-scale precipitation research? I recommend expanding the discussion on this point.

In our work we consider as main result the demonstration, with multi sensor observations, that the spatial and temporal scales of events that produce pluvial flooding in the region are very small even compared with very dense raingauge networks . The fact that a denser monitoring network is needed to catch the rainfall variability scales is a consequence of this evidence. We stated this in the conclusions, rows around 305.

Shen and Xiong (2015) actually used rain gauge information in their study, but with no fusion with radar maps. They found that by increasing the number of stations used over a very wide region in China, the reconstruction of the rainfall field would allow more details of the rain field to be included. However, they used a much larger spatial resolution compared to our study. The paper is now mentioned in the Conclusion section of the manuscript.

Shen Y, Xiong A. Validation and comparison of a new gauge-based precipitation analysis over mainland China[J]. International journal of climatology, 2016, 36(1): 252-265.

2.  Line 88-106: Is this hydrological introduction merely serving as background information? Because it seems that the subsequent results do not involve hydrological models.

    We agree with this comment: to avoid unnecessary details we removed the strictly hydrological part from section 2 as suggested.

3.  Please provide more detail information for DDF curve. And the full name of DDF appears twice in the text, please delete it.

    Additional details of the DDF curves, including the numerical coefficients and the mathematical form adopted, have been added in the text. The full name now appears only once.

4.  Table 2: Please provide the formula expression of $e$ of different return period.

    We indicated by $e$ the percentage relative difference between the mean value of the annual maxima of rain depth at each station and that of the reference station. The mean value has a return period of 2.32 years under the hypothesis of an underlying Gumbel distribution of the annual maxima. The formula expression of $e$ does not depend on the return period and is calculated here for a fixed return period. We slightly modified the text in Table 2 to make it more readable. Also, the word "relative" have been added to the description of $e$ in the text, and the caption of Table 2 was modified.

5.  Line 95: what is a.m.s.l? please provide the full name (above sea level?).

The full text (above mean sea level) is now used.

6. Line 199-201: Many values do not seem to be obtained from this research, please provide citations or Figure num.

   The following reference was added:

   Ferrari, F., Maggioni, E., Perotto, A., Salerno, R., & Giudici, M. (2023). Cascade sensitivity tests to model deep convective systems in complex orography with WRF. Atmospheric Research, 295, 106964.

   and the related sentence was moved to "The Study Area" section.

7. Line 185 – 195: This section also appears to lack citations support.

   Section 4 was removed, and the information therein is now included in sections "The Study area" and "Conclusions". Then the statement concerning the characteristics of the events in the area and their observability with authoritative networks is now a conclusion based on the evidence of our work

8. Line 288: which fusion method is used in this study? Please provide the detail information of the fusion method or citations.

   A description of the fusion method adopted is now included in the text, with appropriate citations.

   Petracca, M.; D'Adderio, L.P.; Porcù, F.; Vulpiani, G.; Sebastianelli, S.; Puca, S. Validation of GPM Dual-Frequency Precipitation Radar (DPR) Rainfall Products over Italy. J. Hydrometeorol. 2018, 19, 907–925

   Pignone, F.; Rebora, N.; Silvestro, F.; Castelli, F. GRISO (Generatore Random di Interpolazioni Spaziali da Osservazioni incerte) –Piogge. Rep 2010, 272, 353

9. Latitude and longitude range is missing in some figures.

   Done

10. It is suggested to indicate the profile direction in Fig.13,14

    Done

This manuscript attempts to study the importance of citizen science on urban rainfall and flooding. While the authors do have interesting results, there are several concerns that need to be addressed before this can be published. Much of these are focused on restructuring the manuscript so that there is a clearer narrative. Major comment 1 summarizes this, but many of my major concerns can be addressed through consideration of if this is helping this study, which to me is answering the question posed on line 286: "What would happen if data from citizen science stations were included in the process of acquisition and production of radar derived rainfall maps?"

Major Comments

1. I believe that the most important portion of this paper is the question that is posed on line 286: "What would happen if data from citizen science stations were included in the process of acquisition and production of radar derived rainfall maps?" It took a long time to get to this point as the manuscript currently stands, and much of the framing done for this paper does not help move this question forward. I appreciate the effort that has gone into the diverse amount of analysis showcased, but I do think that some restructuring needs to be done. For example, moving data descriptions to one central location. Further, one could image a figure where information from Figure 4 is compiled with Figure 11 to further push forward the idea that using more data sources is imperative, especially in urban scales. I think that a careful look at what the authors intended purpose of this paper is, and whether the current structure of the paper serves that purpose. I believe that there is very useful work here, but clarification and a better structure is needed!

   The whole paper was revised with consideration of the comments received from the reviewers, including restructuring the flow of concepts to better highlight the answer to the question indicated. Figures were also revised, and their order changed.

2. The first paragraph in section 3 should be reworked into a more readable form. I think that using something like a "data" section would be useful. Right now, much of the paper is focused on these rainfall gauges, but there were other data sources used. Having these listed in once place would be better. Then, one could just jump directly into the meat of the analysis presented (which I believe is great) starting at line 150. This would also help with removing some of the continuity errors that I am having while reading through the paper (section 5.3 with the information about the sensors that were used).

   The first part of the section has been rephrased to make it more readable. A data section was added (named Data Sources) to describe the characteristics of station data as well as meteorological and remote sensing data sources and access information. This should streamline a bit the description and makes the subsequent sections more readable.

3. As it stands, section 4 doesn't add much to the study. I think that there is a need to contextualize the flooding events that are a result of the rainfall, but this could be better integrated into the analysis of the August 27th and 28th Storm in section 5.

   Section 4 was removed, and the information therein is now included in sections "The Study area" and "Conclusions".

4. As currently presented, you are missing a panel in Figure 7.

   The text was modified to better indicate which radiosoundings are included in the figure (one is not shown).

5. The first mention of any citizen science based platform is on line 255. I would suggest either bringing this forward, reworking some parts of the introduction to be more of a focus on the importance of these data, **or** remove the emphasis of citizen science in the title of this study.

   A data section was added (named Data Sources) to describe the characteristics of station data, including the citizen science stations used in this work.

6. The method of how Figure 12 is important enough to be included and not just referenced. Please add a brief section talking about this.

   A description of the fusion method adopted is now included in the text, with appropriate citations.

   Petracca, M.; D'Adderio, L.P.; Porcù, F.; Vulpiani, G.; Sebastianelli, S.; Puca, S. Validation of GPM Dual-Frequency Precipitation Radar (DPR) Rainfall Products over Italy. J. Hydrometeorol. 2018, 19, 907–925

   Pignone, F.; Rebora, N.; Silvestro, F.; Castelli, F. GRISO (Generatore Random di Interpolazioni Spaziali da Osservazioni incerte) –Piogge. Rep 2010, 272, 353

**Minor Comments (in order throughout the document):**

1. Authors could consider adding the following citations to their outline of sub-daily extremes, especially as these papers are focused on urban areas and heatwaves leading to extreme rainfall (and this seems like a very key point of this study)
   1. Intensification of sub-daily rainfall extremes in a low-rise urban area (https://doi.org/10.1016/j.uclim.2022.101124)
   2. Compound Extreme hourly rainfall preconditioned by heatwaves most likely in mid-latitudes (https://doi.org/10.1016/j.wace.2023.100563)

   Thank you for the suggestions, we included the mentioned citations in the Introduction section.

2. I am unsure if you need the end paragraph of the introduction. I think that there could be a stronger way of ending this section, with a bit more about "to this end, we investigate an intense rainfall event that occurred in the genoa urban center. We present a comprehensive analysis of XYZ…" Etc.

   This paragraph was modified introducing a sentence on the derived merged rainfall maps.

3. Line 69: This could be combined with the paragraph below. It would help tie together that there are complex interactions between the synoptic and regional meteorology, while also pointing out that the complex terrain is a major contribution to rainfall extremes.

   Done.

4. Not a comment in a bad way at all. I love the use of morphological amphitheatre on line 91!

   Thanks for your comment!

5. I think that the hydrologic background is useful, but a bit long. Consider consolidating a bit.

   Following the suggestion some sentences (too much detailed) were removed.

6. Consider showing the locations of the rain gauges on figure 1 if possible. Getting a feel of where these stations are in addition to information in Table 1 would be useful.

   The location of all rain gauges is now shown in Figure 1 and their coordinates are listed in Table 1.

7. On Figure 3&4: instead of the mean and median being a 'thick and thin' line, consider using a dashed like for one to help differentiate.

   The software used is not able to correctly reproduce thin lines, therefore we used the red colour to indicate the mean values.

8. In Table 2 description, please include what is the reference station ( I am assuming it is the GU station in Figure 3).

   The reference station is the DICCA station and suitable indication is now given in the caption.

9. On Figure 5: I suggest demarcating the location of interest to help draw readers attention to the region. You also could consider zooming into the Mediterranean area a bit more.

Done

10. Please consider making the color bar larger on Figures 8 & 9.

Done